# Microbial-Derived Tryptophan Metabolites and Their Role in Neurological Disease: Anthranilic Acid and Anthranilic Acid Derivatives

**DOI:** 10.3390/microorganisms11071825

**Published:** 2023-07-17

**Authors:** Claire Shaw, Matthias Hess, Bart C. Weimer

**Affiliations:** 1Department of Population Health and Reproduction, 100K Pathogen Genome Project, University of California Davis, Davis, CA 95616, USA; clashaw@ucdavis.edu; 2Department of Animal Science, College of Agricultural and Environmental Sciences, University of California Davis, Davis, CA 95616, USA; mhess@ucdavis.edu

**Keywords:** tryptophan, kynurenine, anthranilic acid, microbiome, gut–brain axis, metabolites, neuroactive compounds

## Abstract

The gut microbiome provides the host access to otherwise indigestible nutrients, which are often further metabolized by the microbiome into bioactive components. The gut microbiome can also shift the balance of host-produced compounds, which may alter host health. One precursor to bioactive metabolites is the essential aromatic amino acid tryptophan. Tryptophan is mostly shunted into the kynurenine pathway but is also the primary metabolite for serotonin production and the bacterial indole pathway. Balance between tryptophan-derived bioactive metabolites is crucial for neurological homeostasis and metabolic imbalance can trigger or exacerbate neurological diseases. Alzheimer’s, depression, and schizophrenia have been linked to diverging levels of tryptophan-derived anthranilic, kynurenic, and quinolinic acid. Anthranilic acid from collective microbiome metabolism plays a complex but important role in systemic host health. Although anthranilic acid and its metabolic products are of great importance for host–microbe interaction in neurological health, literature examining the mechanistic relationships between microbial production, host regulation, and neurological diseases is scarce and at times conflicting. This narrative review provides an overview of the current understanding of anthranilic acid’s role in neurological health and disease, with particular focus on the contribution of the gut microbiome, the gut–brain axis, and the involvement of the three major tryptophan pathways.

## 1. Introduction

The metabolome, small molecules produced by the collective metabolism of an organism and its microbiome, is difficult to comprehensively examine, in large part due to the sheer complexity of the interaction between the microbiome and the host metabolic state. This difficulty is especially pronounced in the human gut, where extensive overlap between small molecule metabolism of the host and microbiome exists [1]. Despite this complexity, determining causation and correlation with host health requires distinguishing which molecules are derived from the host’s diet, produced by the microbiome, or produced and secreted into the gastrointestinal tract (GT) from the host metabolism. In the broader context of host health, the importance of the host GT activity and resident microbiome contributions cannot be overstated, specifically with the extensive reach of small molecules to distant tissues in the body [2]. For example, the gut and brain are inextricably linked, exerting bi-directional control, with the gut microbiome playing the part of the complementary brain in this feedback loop [3]. The microbiome density, diversity, and composition differ across the GT. The human GT consists of many specific sections that have differential metabolic capacity and microbiome members, but for the purpose of this review will be broadly discussed in terms of the small intestine, which has low-density microbial consortia and the contrasting microbially dense large intestine [4].

The small intestine utilizes digestive juices and enzymes that are secreted by the pancreas and gallbladder to breakdown carbohydrates, lipids, and proteins. The small intestine also contains microbes, such as *Bacteroides* and *Prevotella*, that aid in the synthesis of micronutrients, like vitamin K2 and B12 [4,5]. The microbiome of the small intestine is exposed to transient conditions that include periodic pH fluctuation from the influx of stomach contents and the microbiome must adapt to the ever-shifting local environment from stomach proximity [6]. This instability may explain why the small intestine harbors only ~10^5^ microbial cells/gram, whereas the large intestine supports ~10^12^ microbial cells/gram [4]. Despite the relatively lower abundance, the small intestine microbiome is an important contributor to host digestion as the small intestine is the primary site of macronutrient digestion and absorption.

Although the small intestine accounts for most macronutrient digestion, the large intestine is an essential contributor to host metabolism and has a differing role in digestion. Host cells of the large intestine, goblet cells and enterocytes, produce mucosa and digestive enzymes for the further breakdown of food matter from the small intestine [7]. The large intestine is also responsible for the absorption of any remaining water and salts not absorbed during transit through the small intestine. In addition to host cell activity, microbes expand the digestive and metabolic functionality of the large intestine by an estimated multiplier of ~150,000 [8], which expands the host’s metabolism beyond its genome capacity.

The microbiome of the large intestine, especially of the colon, is arguably more well studied than that of the small intestine, likely due in large part to the more robust study material, typically feces via non-invasive methods. The diverse microbiome of the colon adds metabolic breadth to the GT, providing the host with access to otherwise inaccessible small molecules and nutrients, like the short-chain fatty acid (SCFA) butyrate, which is a key contributor to healthy gut epithelial function (e.g., increasing tight junction integrity) [9,10] and has shown to alter immune function (for details see Kim [11]). In addition to synthesizing nutrients, the gut microbiome can also produce metabolites, ultimately shifting the balance and activity of overall metabolism.

Amino acids are one substrate set that both host and microbe break down and divert to key and central metabolic pathways. Glutamine and asparagine, for instance, are utilized by host intestinal cells as an energy source [12]. In bacteria, glutamine is also of central importance and serves as a precursor for the synthesis of key nitrogen-compounds and glutamine supplementation has shown to promote and support populations of fiber-degrading bacteria [13,14,15]. The consequential role of the microbiota in amino acid homeostasis has further been highlighted by studies that showed germ-free mice display altered amino acid distribution along the intestinal tract when compared to mice that possessed their natural microbiome [13]. While amino acids are centrally important to protein synthesis and other basic host–microbe functions, amino acids can also regulate host neurological health by serving as precursors to bioactive molecules that circulate throughout the host.

Tryptophan (Trp) is one such amino acid involved in bioactive metabolite production and Trp metabolism is a central pathway shared by the host and its microbiome (Figure 1). Trp is derived from the digestion of dietary protein in the small intestine, where free Trp is absorbed through the intestinal epithelium and enters systemic circulation. Any unabsorbed Trp is transported through the intestinal tract until it reaches the colon where it is utilized by the colonic microbiome [16]. In the colon, Trp plays a major role as a primary substrate for microbial metabolism [17], since easily digested carbohydrates have been utilized early on after indigestion and are in short supply in the distal portion of the digestive system. Trp metabolism is not limited to a single genus of microorganism but is found across many gut microbes, including multiple *Lactobacillus*, *Bifidobacterium*, *Bacteroides*, and *Clostridium* species [18]. Some of the microbes known to utilize Trp are often associated with host health and are part of the gut–brain axis.

Digestion of Trp results in the well-known neuroactive compounds, serotonin and melatonin, and is a starting point for the NAD^+^-producing kynurenine pathway in humans. In addition to the ability to perform parts of the serotonin and kynurenine pathways, microbes have the additional option of producing indoles from Trp catabolism. Humans lack this indole pathway and instead rely on exogenous indoles, either from dietary sources or microbial metabolism, which is metabolically straightforward but the resulting bioactivity in the microbiome is complex and can alter biofilm formation, virulence, and host interaction [19,20]. The kynurenine pathway, one of three main Trp metabolic pathways, has garnered increasing interest for its connection to neurological diseases through the production of the bioactive compounds quinolinic, picolinic, kynurenic, and anthranilic acid in this pathway [21,22,23,24,25,26,27,28]. The three primary Trp pathways are critically interconnected. Shunting Trp into the kynurenine pathway removes the substrate from the production of indole by microbes or serotonin (in microbes and humans). This ‘push–pull’ nature of Trp metabolism, along with the notable bioactivity of the many related metabolic products, make understanding the equilibrium of these interconnected pathways important from the perspective of host health.

The role of microbes as drivers of, and not just responders to, bioactive metabolites (and metabolism) is quickly emerging as an important component of host wellbeing that is modulated by the microbiome co-metabolism. At the same time, this newly elucidated role for microbial metabolites is not well understood or defined. This is in part due to the complex set of in flux metabolites that, combined with microbiome membership changes, leads to altered metabolic potential. Additionally, host diet, the source of substrates for the microbiome, may also be in flux [29]. Many studies examining the gut microbiome focus primarily on taxonomic diversity and do not make the link to metabolism and metabolite changes that are inevitably from diet and host metabolism [30,31]. While a necessary part of the puzzle, bacterial identity alone does not indicate microbiome functionality, which changes the collective metabolism and metabolome [32,33,34,35]. A companion focus on the resultant metabolome is imperative for a holistic understanding of microbiome function, and ultimately, for an understanding of its role in host health. Although there are hundreds of known bioactive metabolites stemming from many dietary substrates, this review will focus first on Trp and then more specifically delve into Trp-derived anthranilic acid and its direct derivatives.

Anthranilate and its derivatives are only one part of the broader Trp puzzle; however, they are an understudied piece with important influence on holistic host health. As will be discussed in more detail later in this review, anthranilate is not often the target metabolite of metabolome/microbiome/host health studies, but a pattern of anthranilate and derivatives is emerging alongside neurological findings. And specifically, as a shared metabolic substrate between host and microbe, a closer look at anthranilate and its derivatives may help fill in some missing pieces around the enigmatic gut–brain and bioactive metabolites connection.

The production, circulation, and interactions of Trp-derived anthranilic acid and subsequent derivatives have profound impacts on host health. Despite the important role of the human microbiome in anthranilic acid production from Trp, no comprehensive review that summarizes the role of microbially derived anthranilic acid in neurodegenerative diseases exists to date. To place anthranilate in the wider framework of this family of bioactive metabolites, this review also contains a brief discussion of the activity of other more widely known neuroactive Trp metabolites. The objective of this review is to highlight the importance of anthranilic acid and its derivatives by outlining the current understanding of anthranilic acid’s role in health and disease, followed by a discussion of the microbes that may affect the production and circulation of anthranilic acid.

## 2. A Gut Feeling: The Microbiome as a Complementary Brain

Microbes in the gastrointestinal (GI) tract work synergistically to break down substrates and survive as a complex consortium with expansive metabolic capability that exceeds human metabolism at least 10-fold [36]. Microbes provide nutrients to the host and it is increasingly clear that the microbiome contributes to host health and behavior through the production of neurologically active compounds, like serotonin, that move from the gut to distant locations throughout the body [37]. Within the gut, these metabolites are often beneficial, supporting immune function and promoting tight junctions, but they can also be detrimental, stimulating inflammation and dysbiosis (Table 1) [38,39,40,41].

Microbial metabolites find their way out of the gastrointestinal tract and into circulation via both passive and active transport [16], and once in circulation bioactive compounds are free to travel to any number of body locations, including the liver, kidneys, lymph nodes, reproductive tract, and even the brain [37,67,68].

Crossing through the gut epithelium into systemic circulation is not always necessary for these microbial metabolites to exert control over host neurology. Interactions between luminal gut metabolites and the host nervous system can be local in nature but with wide-reaching effects [69]. The enteric nervous system, containing sensory afferent and motor efferent nerve fibers, provides a direct link between the brain and physical/chemical gut activity [70]. The Vagus nerve, a part of the enteric nervous system, is central to the basic physiological processes and innervates, among others, the cardiovascular system and respiratory system [71]. The bi-directional (with both afferent and efferent fibers) Vagus nerve is a big part of the gut–brain connection and has recently been shown to exert direct control from the gut to the brain via physical contact with enteroendocrine cells in the gut epithelium [72]. In addition to direct interactions with host organs through active circulation, microbial metabolites can signal to the host via the Vagus nerve and the broader enteric nervous system [73]. Beyond the simple function of digesting food for energy production, the gastrointestinal tract is intimately linked with systemic host physiology through both the uptake and transit of metabolites through the circulatory system and the conveyance of signals through the fibers of the nervous system. This inextricable connection between the host brain and the microbial brain of the gut highlights the importance of metabolic control in host health.

The gut microbiome and metabolome have been correlated with, and in some cases suggested to be, the causative agent in common disorders, such as irritable bowel disease (IBD). IBD is a broad inflammatory disease of the gastrointestinal tract and one of the first conditions that was cited as a link between the gastrointestinal tract and host neurological state [74]. Patients diagnosed with IBD in a Canadian cohort were twice as likely to have mood disorders, such as depression or anxiety when compared to the control group [74]. The directionality of the link between IBD and mood disorders remained ambiguous for many years, as 80% of the IBD cohort were diagnosed with anxiety two or more years before the IBD diagnosis and so directionality nor causation between the mood disorders and IBD was clearly established [74]. A 2016 review re-evaluated this IBD observation in conjunction with other studies to tease apart the gut–brain connection, ultimately finding that mood disorders such as anxiety, depression, and stress all have tight links to increased inflammatory states, potentially contributing to a physiological state that favors a long-term inflammatory disease [75]. Since this initial work in 2008, additional research has emerged on the connection between different gut disease and depression, as well as on the more specific routes and signals of communication between the physically separate body locales of the gut and brain via microbially derived metabolic products [76]. One example is the bacterial-derived metabolite, *p*-cresol, a compound that has been connected to exacerbation of autism spectrum disorders in an expanding body of literature [77,78,79,80,81,82]. *p*-Cresol is an aromatic derivative of tyrosine produced as a catabolite by many members of the gut microbiota, including some *Bifidobacteriaceae*, *Enterobacteriaceae*, *Clostridiaceae*, *Lactobacillaceae*, *Coriobacteriaceae*, and *Bacteroidaceae* [77]. *Clostridia difficile* has displayed a particular growth advantage on *p*-cresol and people with autism spectrum disorders often have a gut microbiome that is enriched in *Clostridia*, suggesting a direct relationship between microbiome membership, microbial metabolism, and neurological symptoms [77]. More systematic studies in mice support this suggestion of a gut–brain axis link for symptoms that fall into the autism spectrum [77,79], further suggesting that microbial metabolism might result in disorders beyond the gut barrier.

Microbially derived metabolic products play a dual function, serving both as substrates for the microbes and as interkingdom signaling molecules between the gut microbiome and peripheral host systems [39,83]. Perhaps one of the most widely recognized examples of these is the mood-modulating serotonin which is a Trp catabolite that acts on multiple brain regions via its specific receptors that are found throughout the circulatory system, but also in the brain [84]. Though known primarily for its activity in the brain, over 90% of serotonin is found in the GT lumen [85] (Figure 1). Other small molecules, such as SCFAs are imperative for healthy gut function, but they are also able to cross the gut epithelium and have been reported to breach the blood–brain barrier in small concentrations [86]. SCFAs that make the journey to the blood–brain barrier help to restore the barrier’s integrity [86]. These small molecules can serve as sources of energy (e.g., succinic acid) for the host and multiple members of the microbiome, but their role as signaling molecules with an array of effects should not be ignored due to their wide array of functional activities [87,88]. Propionic acid, an SCFA produced through fiber and amino acid fermentation by gut microbes in the colon, is transported through the portal vein to the liver, where it can inhibit liver damage and [89] gluconeogenesis [90]. Propionic acid is a microbially derived metabolite with broad implications for host health given its ability to move to the liver and into circulation, where the compound can exert effects across a range of distant sites to the GT. Microbial metabolites that cross the epithelial barrier or interact with the host nervous system from the GT have the potential to act upon several tissue systems ranging from the circulatory system to the brain [91]. The system-wide health effects of microbial metabolites in the host makes the modulation of these pathways a relevant strategy that might alter the outcome of many human diseases. Microbial metabolism can drive the production of bioactive metabolites that have either direct roles in the development and progression of diseases, or that exert more indirect effects on the disease progression. Modulating these microbial metabolite–host system interactions, like those seen with metabolites from Trp catabolism, through microbial engineering, gut microbiome remodeling, or dietary interventions could produce treatment for diseases for which effective treatments are currently not available or are instead extremely invasive.

## 3. The Tryptophan Pathway: Microbial-Derived Metabolic Mind Control

Trp is a significant contributor to the crosstalk between the gut microbiome and the brain. As the precursor for many important products, like indole, serotonin, and NAD^+^, Trp is crucial to basic microbial and human physiology. Trp is an essential amino acid that is introduced to the host primarily through diet, though some gut microbes can synthesize Trp from indole and serine via the Trp synthase enzyme complex (TrpAB) [83,92]. A study looking at the distribution of amino acid biosynthesis pathways across the publicly accessible genomes of 2856 human GT microbiome members from over 800 distinct microbial species found Trp biosynthesis was highly variable and individual microbes lacked a complete enzymatic repertoire to produce Trp [92]. There was a group of 38 genomes that predicted Trp auxotrophs making them unable to produce anthranilate, but interestingly possessed the genes for synthesis of Trp from anthranilate [92]. The authors posited that Trp biosynthesis pathways may show high variation across human gut microbes due to the high energetic cost of producing Trp in comparison to other amino acids, because Trp is encoded by only a single codon, UGG, and is a rarely incorporated amino acid with high molecular weight (204.23 g/mol) [92]. The high variation in Trp biosynthesis across gut microbes pulls the focus from basic host use towards the microbial use of this central amino acid and the metabolites resulting from such use. Trp, regardless of the source, is central to many cellular functions and the general path from Trp to bioactive derivatives is an important study for host health and disease impacts.

Trp in the host intestinal tract is freed during protein digestion within the small intestine or via microbial digestion and biosynthesis in the large intestine. Free Trp in the small intestine is absorbed into systemic circulation by the transmembrane broad spectrum neutral amino acid transporter (B^0^AT1) that is encoded by the *SLC6A19* gene in intestinal epithelial cells [93]. Interestingly, 19 *SLC6A19* paralogues that encode transporters in the GT have been reported. All paralogues are part of the SLC6 transporter family, a group that is implicated in human diseases and are actively being investigated as therapeutic targets for diseases that include Parkinson’s, depression, anxiety, hyperekplexia, and Tourette Syndrome [93,94]. Once in systemic circulation, albumin-bound Trp is carried to the liver where it is digested via the kynurenine pathway, or to the brain, where Trp crosses the blood–brain barrier with the help of amino acid transporters and is shunted into both the serotonin and kynurenine pathways [95,96]. Neutral amino acid transporters able to facilitate Trp-derivative absorption exist in the small intestine, but most of the gut’s Trp metabolism occurs in the large intestine and is mediated by microbial metabolism [97]. There is some evidence that disease states, such as small intestinal bacterial overgrowth (SIBO), can shift Trp metabolism in the small intestine, driving an increase in locally produced kynurenine [98], but the basal level of kynurenine production in the small intestine appears to be low under normal in vivo conditions [53]. The fraction of Trp that is not digested and absorbed in the small intestine is instead digested by the gut microbiome of the large intestine via the kynurenine, serotonin, and indole pathways [16]. Trp metabolites produced via microbial metabolism in the large intestine are utilized locally or cross the GT epithelium via passive diffusion (kynurenine and indole) or with the help of transporters (serotonin) [99,100]. Bioactive molecules stemming from Trp digestion, whether from digestion in the small intestine or from microbial production in the large intestine, can act on the local gut tissue or exert control over distant host tissues via circulation.

The bioactive products of these pathways, like kynurenic acid, have been associated with neurological conditions. Alzheimer’s, schizophrenia, and depression have been linked to the serotonin, kynurenine, and indole pathways, and intermediates of these pathways have been identified as potential causative agents or as potential biomarkers for disease states [21,23]. The microbiome, alongside the host, contributes to fluctuating intestine luminal and circulating serum levels of Trp-derived bioactive compounds, which are critical to the regulation of the gut–brain axis. A more mechanistic understanding of the function of these compounds in disease and of the microbial control of their production is imperative for improved neurology therapeutics. The three major pathways of Trp digestion, microbial metabolism, and the link to neurological diseases will be discussed briefly below and are summarized in Figure 2.

### 3.1. Indole

Exogeneous Trp that makes it through the small intestine and into the large intestine is primarily digested by gut microbes, as colonic epithelial cells do not absorb amino acids in great quantity [101]. Whereas gut microbes can convert Trp into indole via a one-step reaction catalyzed by tryptophanase (TnaA), humans lack the ability to synthesize indole and depend instead on the microbially derived indole. Indole and its directly related compounds, collectively called indoles, include indole-3-acetamine, tryptamine, indole-3-acetylaldehyde, and inole-3-pyruvate, and are produced by members of the *Proteobacteria*, *Firmicutes*, *Fusobacteria*, and *Bacteroidetes* [102]. Further transformation of indoles results in the production of indole-3-acetate, indole-3-aldehyde, and skatole [102]. Not all microbes can produce indole. Fungi and bacteria that are unable to synthesize indole or its derivatives still have mechanisms for sensing and utilizing indoles as metabolic precursors or ligands for signaling cascades [103,104]. *Candida albicans*, *Staphylococcus aureus*, and *Pseudomonas aeruginosa* all use indoles to induce biofilm formation, and *Salmonella enterica* serovar Typhimurium relies on indole signaling to activate an oxidative stress response to resist antibiotic activity [104]. Indoles that are not utilized by other gut microbes and instead absorbed into circulation can be used in the liver to produce indoxyl sulfate, a toxic metabolite known for its ability to affect drug clearance in the liver through the modulation of drug transporter expression [105]. Indoles also act as interkingdom signaling molecules between microbes and their host, promoting tight epithelial junctions and regulating inflammation [39,106]. Indoles can directly induce interferon and interleukin 22, two immune system products that mediate repair of the gut barrier and inflammation, but this induction appears dependent on acute stressors in the gut and is thought to be a spatially regulated response [106]. Indoles regulate a variety of bacterial responses within the gut microbiome itself, including biofilm formation and virulence [19,107,108,109,110].

Humans may not be able to produce indoles, but they do interact with indoles via the gut epithelium and through systemic circulation. Indole concentrations in feces of healthy individuals range widely, and studies suggest there are many factors contributing to this variance, including microbiome diversity, free exogenous Trp, and the availability of other substrates [103,111]. Dietary Trp is also a significant driver of luminal indole concentration [112]. Once dietary Trp is microbially converted to indoles, it can travel through the lumen and gut mucosa via the host’s aryl hydrocarbon receptors (AhR) on the gut epithelial cell surface [113]. AhRs are not limited to the gastrointestinal tract, instead this cytosolic receptor is broadly distributed in the host [105,114,115]. AhRs are most highly expressed in the nervous system, especially the tibial nerve, and in the lungs, but are also expressed across much of the cardiovascular system and in reproductive tissues as well [116]. The breadth of tissues expressing these receptors is due to the crucial role of AhRs throughout the human body. AhRs serve as a mediating signal between environmental conditions and appropriate host response, allowing for dynamic interplay between internal host physiology and fluctuating environmental cues [117]. As a signaling mechanism, AhRs are a key link between the gut microbiota and host via their binding to numerous ligands [118], but these receptors and indole together display a unique affinity for one another [113]. Interestingly, though also found in mice, AhRs and indole in the mouse gut do not display the same unique binding activity as has been observed in human cells [113]. In humans, once bound with indole, AhRs move intracellularly and, through modifications within the cell, regulate the associated target genes [119].

Though indole is now understood to be a central player in the metabolic crosstalk between host and microbiome, current research primarily focuses on the localized effects in the GT system, neglecting potential effects of indole on distant tissues. Indole can affect host neurology from the gut lumen via the enteric nervous system, or through direct contact with host tissues via transit in the circulatory system. In the lumen, indole can activate enteroendocrine L-cells, which are specialized and electrically excitable epithelial cells that interface with the gut lumen, secrete peptide hormone glucagon-like peptide1, and are directly innervated by enteric afferents [120]. In addition to stimulating nerve-connected L-cells, indole can also cross the blood–brain barrier, granting microbially derived metabolites direct access to host neural circuitry inside the brain [121]. In a recent study using a rat model, the effect of indole derivatives oxindole and isatin on depressive and anxiety-related actions was examined [122]. Rats injected with an acute high dose of these indole derivatives in the cecum displayed lower motor activity and increased anxious behaviors [122]. To test the effect of a less extreme condition, gnotobiotic rats were inoculated with indole-producing *E. coli* to generate more constant and medial indole levels [122]. Rats with indole-producing bacteria were found to display more anxiety behaviors as compared to their uninoculated counterparts [122], suggesting that even modest concentrations of elevated indole promote anxiety.

Somewhat contrasting to the behavioral effects observed by Jaglin et al. (2018), Wei et al. (2021) reported that indole promoted neurogenesis of adult brain tissue in mice [123]. Mice treated with indole, both through systemic inoculation and via monocolonization with *E. coli*, showed increased neuronal growth and greater circuitry development than germ-free mice without access to indole [123]. Indole’s part in reviving neurogenesis in the adult mouse brain, where neural growth is typically on the decline, is linked back to AhR-mediated signaling and the upregulation of β-catenin, Neurog2, and VEGF-alpha in the hippocampus [123]. The results of this study, which investigated both mechanistic and holistic actions, underscore the importance of microbial metabolites in regulating host neurology.

### 3.2. Serotonin

In contrast to indole, serotonin can be synthesized both by gut microbes and by the host in various tissues. Serotonin (5-hydroxytryptamine) and melatonin (N-acetyl-5-methoxytryptamine) are well known as mood- and sleep-regulating neurotransmitters. Though a key element in neural activity, only 5% of the body’s serotonin is found in the brain, the remaining 95% of serotonin can be found instead in the gastrointestinal tract, primarily in serotonin-producing enterochromaffin cells of the gut barrier [85]. Serotonin is also the precursor of melatonin, which is synthesized in both humans and microbes via a two-step acetylation and methylation process [124]. Studies with gnotobiotic mice indicated that although gut serotonin is produced mostly by host cells, the microbiome still plays a crucial role in the regulation of serotonin production, both via the consumption or production of metabolic precursors and through control of host activity by signaling cascades [125].

Human production of serotonin in the gut primarily takes place in the epithelial enterochromaffin cells via the activity of the rate-limiting Trp hydroxylase enzyme (TPH1). Although the serotonin-producing enterochromaffin cells of germ-free mice are normally developed, these germ-free mice are deficient in serotonin and display a Trp buildup in the large intestine [126]. In conjunction with decreased serotonin production, the expression of serotonin transporter SLC6A4 is increased, likely to compensate for the diminished levels. This same trend of decreased serotonin and increased Trp is not observed in the inherently microbially sparse small intestine [126]. To explain this microbial-induced serotonin deficit, one study inoculated gnotobiotic mice with altered Schaedler flora, a defined consortia of gut microbes that is widely used with mice models. Interestingly this study found that the deficit remained even in these minimally colonized mice [126]. Colonic serotonin levels did increase back to normal levels after inoculation with spore-forming members of the specific pathogen-free microbiota, mostly clostridial species, via stimulation of *TPH1* expression in enterochromaffin cells [126]. Gnotobiotic mice studies have demonstrated that the microbiome is responsible for modulating around 64% of the colonic serotonin concentration and affected almost half (49%) of the circulating serotonin [126,127]. It appears that the microbial control of serotonin production by the host is not a one-way mechanism, as gut bacteria have also been shown to modulate their activity in response to colonic serotonin concentrations. Increased luminal serotonin levels stimulate bacterial synthesis of serotonin [126] and serotonin produced in the GT improves the local integrity of the enteric nervous system, which then regulates gut motility and nutrient absorption [128]. Circulating serotonin has been reported to be excluded from crossing the blood–brain barrier (BBB) due to its slight polarity and is instead produced in the brain by the raphe nuclei from free Trp, which does cross the BBB [129]. It is noteworthy that BBB permeability is altered by many factors including age, temperature, and pharmaceuticals as extensively reviewed by Zhao, Gan [130]. Pathogens can also alter BBB permeability or take advantage of decreased barrier integrity, as seen with uropathogenic *Escherichia coli* reducing activity in the TGFBRII/Gli-2 signaling pathway that normally works to maintain barrier integrity [131]. Changes in the BBB integrity open the door for normally blocked substrates, like serotonin and other small molecules, to cross this barrier and gain access to the brain. The reported inability of serotonin to cross the BBB suggests that high levels of serotonin produced in the GT has a function outside of regulating behavior via direct interaction with receptors located in the brain. Indeed, serotonin is a potent and ubiquitous neurotransmitter involved in the pulmonary, enteric, immune, and circulatory systems that remains to be studied for the exact linkage from the GT to the brain and the possible behavior changes that this compound can impart [132,133,134].

Melatonin, like serotonin, can be produced by the host and when secreted by the pineal gland in the brain, melatonin exhibits behavioral control. Host melatonin production in the brain typically follows an exogenous light cycle to sync host behavior to light exposure [135]. Interestingly, it is also produced by enterochromaffin cells in the gut at concentrations 400 times higher than what has been observed for pineal gland production [136]. In the gut, melatonin contributes to the relaxation of gut tissue and lower overall motility, effects which oppose those of serotonin [137]. Exogenous melatonin administration alleviated some irritable bowel (IBS) symptoms, potentially due to the relaxation regulation by melatonin [138], but further studies are needed to pinpoint a specific mode of action. Additionally, melatonin supplementation was shown to reduce obesity-related bacterial taxa in mice on a high-fat diet, restoring the microbiota to that resembling the one associated with normal chow mice [139]. Melatonin may also be a potential therapeutic agent in neurological disorders like multiple sclerosis and Huntington’s disease [140], but more work in this area is needed to understand the mechanics of this potential metabolite as a therapeutic. The regulation of host and microbe activity by gut-derived serotonin and melatonin is a crucial aspect of the gut–brain connection. Though these bioactive Trp metabolites may be well known, they are not necessarily well understood. Further evaluation of how serotonin and melatonin regulation may play into the control of neurological disease will be critical for the development of advanced disease treatments.

One aspect of melatonin that also requires further study is its role as a signaling molecule amongst gut microbes. Multiple studies have reported the use of indole as a mechanism for bacterial communication, allowing both for symbiotic coordination amongst gut bacteria and conveying the existence of competitive environmental conditions [19,141,142]. Gut-derived melatonin may also support chrono-related bacterial changes. *Klebsiella* (formerly *Enterobacter*) *aerogenes*, a human commensal organism, shows increased motility with increased luminal melatonin levels, indicating that the host secretion may be regulating the microbial activity in a circadian-dependent manner [143]. In vitro work provided evidence that *K. aerogenes* regulates key physiological pathways in response to melatonin exposure, including pathways involved in stress response, carbohydrate transport, and metal ion homeostasis [144]. This same study also showed circadian rhythm-dependent synchronization of gene expression across *K. aerogenes* cultures in response to melatonin concentration [144]. Though this same depth of study regarding the gene expression and growth response of a single gut commensal to melatonin has not been conducted in many other microbes, it stands to reason that the circadian clock and sensitivity to melatonin is not unique to *K. aerogenes*. The exploration of circadian rhythmicity in gut bacteria may provide an even more direct connection between host homeostatic rhythms and microbiome activity, further tightening the link between hosts and their microbes.

### 3.3. Kynurenine

Utilizing approximately 90% of Trp, the kynurenine pathway is a robust multistep metabolic pathway with numerous bioactive intermediates and the terminal product of NAD^+^, as well as other small molecules [145]. Kynurenine metabolism produces both neuroactive and neurotoxic products which creates a ‘push–pull’ network that depends on other aromatic compound intermediates in the overall pathway. This complex metabolic pathway has garnered interest for its many bioactive metabolites and for its role in neurological diseases [146,147].

The production of kynurenic acid, a single-step metabolite of kynurenine, pulls the pathway flux from the production of neurotoxic quinolinic acid; thereby detoxifying the local niche. Due in part to this metabolic trade-off, kynurenic acid is thought to have some neuroprotective properties [148]. In a study looking at metabolic markers of fibromyalgia and chronic fatigue syndrome, the kynurenic acid/quinolinic acid ratio decreased in patients with chronic fatigue syndrome [148] and kynurenic acid/3-hydroxykynurenine ratios decreased in patients with fibromyalgia compared to healthy patients [148]. These findings suggest these ratios could serve as potential biomarkers to evaluate the overall metabolite balance of patients. A different study, investigating kynurenine levels in the blood of dementia patients over a duration of 5 years, found that both unusually high and low levels of circulating kynurenine correlated to negative cognition prognoses [149]. During this work, the investigators looked more holistically at the pathway and related metabolic ratios and found that mean kynurenine serum levels did not appear to correlate with progressing dementia [149]. Instead, a non-linear relationship was detected where deviation from midline kynurenine concentrations in either direction was associated with more extreme neurological and dementia symptoms [149]. Kynurenine metabolites have also been linked to bipolar disorder, which is a complex neurological disorder that affects approximately 2% of the population [150,151]. Notably, recent meta-analyses have shown there is a connection between metabolites of the kynurenine pathway and bipolar disorder, but the mechanistic and possibly causative relationship remains less well understood [150,151]. Though the causative relationship may not yet be fully elucidated, it is nevertheless clear that metabolites stemming from the kynurenine pathway play key roles as markers of disease or perhaps even causative agents.

Though current scientific literature has abundant coverage of the widespread implications and potential systemic results of the dysregulation of the kynurenine pathway, a deep dive into the many branches of this pathway is outside the scope of this review. Fortunately, there are many excellent reviews on this subject [152,153,154,155,156]. Instead, the remainder of this review will focus solely on one kynurenine metabolite that may hold the key to understanding a subset of neurological disease in the host–microbe context: anthranilic acid.

## 4. Anthranilic Acid and Beyond

Anthranilic acid, a kynurenine metabolite produced by host and microbe alike, is a bioactive compound with potential systemic neurological effects. Though in the past much research has focused on the kynurenine pathway at large, the role of anthranilic acid in the gut–brain axis is emerging as an important piece of the puzzle. Anthranilic acid is a direct metabolic product of kynurenine digestion in humans and microbes and represents an alternate branch to two other immediate products (i.e., 3-hydroxykynurenine and kynurenic acid) [157]. Kynurenine is hydrolyzed to anthranilic acid and L-alanine with the help of kynureninase. With the transfer of an amino group, kynurenine becomes the alternate product kynurenic acid, and the addition of an oxygen by kynurenine-3-monoxygenase turns kynurenine into 3-hydroxykynurenine. Anthranilic acid, through non-specific hydroxylation, becomes 3-hydroxyanthranilic acid, which is the precursor for quinolinic acid, picolinic acid, and the terminal product NAD^+^ [157]. The ratio of anthranilic acid to the downstream 3-hydroxyanthranilic acid has been suggested as one potential biomarker for both neurological and physiological disorders [55]. Given its pivotal position as a diverging branch of the kynurenine pathway, its bioactive properties, and the dual production by host and microbes, anthranilic acid is a key Trp metabolite to study in the context of the gut–brain axis.

Pharmaceutical research has built on the inherent production and utilization of anthranilic acid metabolites in vivo for the creation of novel drugs and therapeutics. A 2021 review of anthranilic acid medicinal research highlights the bioactivity of anthranilic acid derivatives [158]. In this review, anthranilic acid is discussed as a pharmacore, which is a basic compound that is biologically active and that provides a starting point for more complex pharmaceuticals. Specific substitutions and additions on the basic anthranilic structure, 2-amino benzoic acid, yielded active therapeutics with a wide array of targets, and anthranilic acid analogues are currently in use for the treatment of multiple metabolic disorders and as anti-inflammatories. Analogues created in pharmaceutical research have exhibited anti-viral properties, activity on drug-resistant cancer cells, and some varied effects on basic cellular processes like the hedgehog signaling pathway [158]. Other reviews and research in this area have likewise supported the notion of anthranilic acid as a key contributor to pharmaceutical research, due in large part to the highly active role anthranilic acid already has in vivo [159,160].

As noted above in the context of pharmaceuticals and further outlined in the sections below, anthranilic acid and chemically similar compounds are important in host physiology beyond the localized intestinal tract. Dual production of these compounds by the host and the microbiome increases the complexity of this host–metabolite interaction. A more mechanistic understanding of anthranilic acid as an agent of neurological disease is imperative.

### 4.1. Human Production of Anthranilic Acid and Derivatives

The production of anthranilic acid by humans is integral to this review since it supports the concept of co-production between the host and the microbiome of a bioactive compound. Though the gut microbiome contributes many important and biologically relevant metabolites, humans make their own contributions to systemic metabolism. Many host tissues and cell types, including neurons and macrophages, have the enzymatic repertoire to support most or all of the kynurenine pathway [161,162,163]. There are various drivers of the kynurenine pathway in vivo and many factors that exert control over the divergence into the many branches and resultant metabolites.

In vitro studies have shown the kynurenine pathway in multiple human cell types can be induced and specifically altered by interferon-gamma [164]. Macrophages in particular were driven towards the production of 3-hydroxyanthranilic acid from L-tryptophan via interferon-gamma activation [164]. A recent review has further highlighted the interplay of immune markers and regulation of the kynurenine pathway, stating that key to this relationship is the indoleamine 2,3-dioxygenase enzyme (IDO) [165]. IDO appears to participate in crosstalk with markers of inflammation and though IDO plays a secondary role in pushing TRP towards kynurenine and further down, kynurenine-metabolites under homeostatic conditions, the onset of inflammation drives IDO activity [165]. Pro-inflammatory cytokines, IL6, TNF-alpha, and IL-4, drive the expression of IDO through shared signaling pathways, like through the activation of AHRs [166]. IDO expression creates an environment that suppresses the immune response through the production of neurotoxic metabolites and depletion of the central immune-mediating amino acid Trp [167]. Multiple studies have also revealed that IDO is highly induced in colonic tissue and mucosa of patients with IBD and Crohn’s diseases, both diseases that are marked by chronic inflammatory states in the gut and potential microbial interference, but for which the underlying taxa and mechanisms are currently unknown [168,169]. Inflammation is often correlated, if not implicated, with neurological disorders and declining health. The tight partnership between IDO activity and inflammatory markers may provide one piece of the metabolic–neurologic activity relationship puzzle. Tryptophan 2,3-dioxygenase (TDO), like IDO, is a rate-limiting enzyme that controls conversion of Trp into the early metabolites of the kynurenine pathway [18]. Whereas IDO can be found throughout host tissues, it is primarily expressed in the liver, where 90% of kynurenine metabolites are produced [170]. Hormonal cascades in the host, such as those induced by stress, have been shown to drive TDO activity in the liver and cause a resultant increase in circulating kynurenine metabolites [170].

Hepatic circulation and utilization of Trp metabolites are not limited to the kynurenine pathway. Tryptamine, produced from the decarboxylation of Trp by gut microbes, attenuates pro-inflammatory cytokine production in the liver [171]. Skatole is another Trp derivative, specifically a product of the indole pathway, that is produced in the gut by microbes but potentially bioactive in the liver. One study found that patients without liver disease (hepatic encephalopathy) had no detectable serum skatole, but those with signs of liver disease had detectable skatole levels between 0 and 442 nmol/L [172]. This study did not test the directionality of skatole levels and so did not postulate whether skatole was a causative agent of disease or a byproduct [172]. Skatole may be toxic at high concentrations in the body, but a recent study showed skatole may attenuate hyperlipidemic conditions of the liver [173]. Trp metabolites produced by microbes in the gut, like skatole and tryptamine, find their way to the liver through hepatic circulation and once localized, can either attenuate or exacerbate disease conditions. *p*-Cresol, a product of amino acid fermentation and [2] mentioned previously in relation to autism spectrum disorders, is another metabolite that is derived from microbial activity that enters hepatic circulation. Modification of *p*-cresol in the liver gives rise to uremic toxins and can intensify renal disease as well as promote systemic inflammation [2]. Microbial compounds in the gut that make their way to the liver through hepatic circulation can then be further modified by the host, altering catabolite activity. Though liver activity is integral to understanding the interaction of Trp metabolites and host health, the gut–liver axis is not the direct focus of this review. More in-depth coverage of the gut–liver axis and the contributions of gut microbes can be found in other reviews [2,174,175].

There are internal controls of metabolic-related enzymatic activity in the liver, and the original limiting factor for the kynurenine pathway is ultimately the availability of the primary substrate, Trp. As Trp is an essential amino acid only acquired from diet and microbial metabolism, the availability of Trp in the gut is a necessary starting point for the estimation of circulating secondary metabolites. An 8-week-long high-fat diet in mice resulted in a measurable perturbation of gut taxa and subsequently significantly altered the gut, serum, and liver metabolome [171]. The gut microbial community has been shown to be a key driver of metabolism and so it cannot be discounted in the discussion around circulating kynurenine metabolites like anthranilic acid. Despite our understanding of the human contribution to circulating anthranilic acid compounds, it is not always possible to disentangle host and microbe metabolisms from one another. The inability to fully dissociate the production of anthranilic acid complicates the already challenging problem of tracing this metabolite’s biological activity in the host. Just as is it important to contextualize anthranilic acid research with the potential metabolic contributions of the humans being studied, so too is it important to understand microbial contributions to the shared metabolic pool.

### 4.2. Bacterial Production of Anthranilic Acid and Derivatives

The gut microbiome has the functional capacity to produce anthranilic acid and downstream metabolites from the breakdown of kynurenine, and ultimately from the original source of Trp. Research on microbial production of anthranilic acid and its derivatives mostly focuses on industrially applicable microbes, which produce anthranilic acid in large quantities Anthranilic acid, and its derivatives, are used in the pharmaceutical industry for a myriad of applications, including as antimicrobials and as a potential regulator of diabetic symptoms [158]. In the food industry, the anthranilic acid derivative, methyl-anthranilate, is used as a flavor compound to impart a grape essence to candies and other food items [176]. Many microbes, though it may vary by species, have the enzymatic repertoire to perform either all or part of the kynurenine pathway [177]. A search using the OrthoDB database revealed six major bacterial phyla contain organisms with the kynureninase gene (*KYNU*), which catalyzes the conversion of kynurenine to anthranilic acid (Figure 3) [178]. Anthranilic acid to 3-hydroxyanthranilic acid is a non-specific hydroxylation step, but the conversion of 3-hydroxyanthranilic acid to subsequent quinolinic acid relies on the expression of 3-hydroxyanthranilate 3,4-diozygenase (*HAAO*). Five major phyla contain organisms with the *HAAO* gene, including the gut-dominating *Firmicutes* and *Bacteroidetes* (Figure 3) [178]. The microbial production of anthranilic compounds in the gut is an important, though understudied, area of research on Trp-related neuroactive compounds.

Studies that focus on the production of anthranilic acid by gut microbiota many times rely on a more universal approach to this analysis, highlighting the production in the gut by broad functional groupings and not by a specific taxon. The difficulty of teasing apart microbes and their metabolic products in the gut is in large part due to the complex interactions between microbes and the present diversity [177], which is coupled to a very high genomic diversity of microbiome members [179,180]. One point that is indisputable, however, is that the gut microbiota plays a major role in shifting the availability of free Trp within the host. Germ-free mice have altered free Trp levels compared to inoculated mice and show increased kynurenine pathway metabolites [181,182]. The gut microbiota at large has the capacity to support the kynurenine pathway in the gut. Previous work demonstrated that the genomes of gut bacteria contain the necessary enzyme homologs to produce anthranilic acid and kynureninase, as well as other enzymes responsible for producing neuroactive metabolites in this pathway [97,177]. The strain or species that have the higher or lower metabolic capacity for Trp turnover are largely missing from the literature in this area.

The ability to produce anthranilic acid or other Trp-related neuroactive metabolites may be species specific. Metabolic production is also context specific, as the host dietary intake and the resulting concentrations of these shared metabolic products shifts the equilibrium for both microbe and host. Human pathogen *Psuedomonas* can produce multiple components of the kynurenine pathway [49]. More relevant to this review, however, is that *P. aeruginosa* produces anthranilate and then uses it as a precursor to several virulence factors [49]. Kurnasov and colleagues characterized the anthranilate pathway in prokaryotes by conducting a genome comparison and searching for three enzymes, tryptophan 2,3-dioxygenase, kynurenine formamidase, and kynureninase, which together produce a eukaryotic-like anthranilate pathway [43]. Among others, *Burkholderia fungorum*, *Bacillus cereus*, *Ralstonia solanacearum*, and *Ralstonia metallidurans* were found to encode for all three anthranilate-related enzymes [43]. Another study looking at cold stress in pigs and its effect on microbiome diversity and activity identified anthranilic acid as one of the few metabolites that increased under cold stress conditions [183]. *Prevotella* UCG-003 was revealed to be the most strongly correlated microbe with this cold stress-induced increase in anthranilic acid [183]. It is well defined that the gut microbiota is an important contributor to the production of neuroactive compounds from Trp.

### 4.3. Neurological Outcomes of Anthranilic Acid

Research on the mechanistic role of anthranilic acid in neurological diseases has been limited up until now. Many studies focus primarily on the use of anthranilic acid and its derivatives as biomarkers [184], but do not necessarily drill down to determine the causative or protective effects of these compounds on host systems. While there is a place for studying these bioactive metabolites from the perspective of diagnostic potential, future therapeutic development will inevitably rest squarely on a mechanistic understanding of metabolic bioactivity. Current literature suggests that anthranilic acid leads a double life; the metabolite appears to exert neuroprotective effects in some cases while displaying neurotoxic effects in others [185,186]. The potential neurological activity of anthranilic acid in vivo is complex and at times conflicting. Anthranilic acid resides at the increasingly explored intersection of diet, microbe, and host neuroactivity, and a more thorough understanding of anthranilic acid’s bioactivity is important for future use as a biomarker or therapeutic target.

Anthranilic acid is often seen as part of the neurotoxic branch of the kynurenine pathway, a branch that includes quinolinic acid and hydroxykynurenine [187,188]. Elevated serum levels of anthranilic acid have been measured in people with schizophrenia and in those diagnosed with Parkinson’s disease [184,189,190]. Oxenkrug and colleagues showed circulating anthranilic acid levels in patients with schizophrenia were two times higher than those of the control population, confirming the results of a previous rat model study [184]. Though the study could not attribute the elevated anthranilic acid to anything specific, such as increased enzymatic activity in the pathway, the authors did affirm that the increase was not correlated with anti-psychotic drug intake [184]. Adding another layer of complexity to this puzzle is the difference seen between the sexes. One study found that increased circulating anthranilic acid in schizophrenic subjects was not only related to neurological diagnosis, but also correlated with being female [189]. Females in the study had plasma anthranilic acid levels 27% greater than those of their male counterparts, all of whom had a schizophrenic diagnosis [189]. It is possible this difference could be explained by sex hormones. For example, estrogen can inhibit some enzymes of the kynurenine pathway, or potentially by the deficiency of these enzymes resulting from other multifaceted signaling cascades that differ between biologically male and female subjects [191,192]. The impact of estrogen on other signaling pathways has already been shown with another host and microbe-derived neurotransmitter, dopamine. Previous work established estrogen affects dopamine activity, typically boosting synthesis and release, in both animal models and follow-up human studies [193]. Gut bacteria, including *E. coli*, some *Bacillus* species, *Klebsiella pneuomoniae*, and *Staphylococcus aureus*, have been reported to produce dopamine from tyrosine [194]. Though primarily thought of as a mammal’s molecule, dopamine may be produced by bacteria as a quorum-sensing molecule, increasing motility and biofilm formation, or as a growth-enhancing factor, through improved iron regulation [194]. The interplay between the hormone estrogen, neurotransmitter dopamine, and microbial production of these bioactive compounds is already established in the literature, so it is not surprising to see emerging literature suggesting a hormone–metabolite crosstalk in the context of other microbially produced bioactive metabolites like kynurenine.

Beyond schizophrenia, other neurological conditions (e.g., chronic migraines) have been correlated to the levels of circulating anthranilic acid. A study on chronic migraines in humans tracked the serum levels of multiple kynurenine metabolites and found a 339% increase in the concentration of anthranilic acid in those suffering from chronic migraines [195]. In tandem with that increase, the same study showed a 63% decrease in the downstream metabolite 3-hydroxyanthranilic acid [195]. Importantly, the study excluded subjects with neurological comorbidities, suggesting that this marked an increase in anthranilic acid was specific to migraines and not due to a known underlying cause. It is possible that the raised anthranilic acid level is a response to neural activity and simply marks the diagnosis of chronic migraines. More interestingly, the authors of this study propose a combination of N-methyl D-aspartate (NMDA) receptor overactivity in migraines and the reduction of neuroprotectant kynurenic acid, which correlates with an increase in anthranilic acid, which together contribute to migraine symptoms [195]. This same research group also demonstrated concurring results in patients with a similar diagnosis of repeated cluster headaches. Those with cluster headaches had a 54% increase in serum anthranilic acid and a 54% decrease in 3-hydroxyanthranilic acid as compared to the healthy control patients [196]. Taken together, these studies indicate anthranilic acid is certainly a part of headache-related diagnoses. It remains to be determined if this metabolite is a causative or a responsive biomarker.

Evaluation of depression’s metabolic roots has indicated that there are correlating Trp metabolite concentrations. In one study on circulating anthranilic acid in patients diagnosed with depression, anthranilic acid was found to be inversely correlated to the severity of depression, though this finding was dependent on sex [197]. As was indicated in the case of schizophrenia and biological sex, anthranilic acid levels associated with depression appeared more pronounced in women as compared to men in the same study [197]. This study was careful to exclude medication as a factor. Previous work has noted a difference in circulating kynurenine metabolites between medicated and non-medicated subjects [198], making medication an additional confounding factor for gut–brain axis research. A study detailing the onset of depression in patients undergoing treatment for Hepatitis C showed a decrease in Trp availability with a corresponding increase in anthranilic acid circulation [199]. During this 24-week longitudinal study, subjects with increased anthranilic acid levels were significantly more likely to be diagnosed with major depressive disorder [199]. The role of anthranilic acid as a biomarker, or even as a cause, of depression disorders is an ongoing area of research with promising findings for the development of potentially more targeted treatments. There are, however, conflicting studies which show no correlation or weak correlation between depressive symptoms and anthranilic acid serum concentrations [200]. As human studies are often rife with confounding factors, this conflict amongst studies comes as no surprise. Biological sex [189,197] and medication [198] can affect the availability of Trp metabolites and are examples of factors that are not always accounted for but can play significant roles in determining the outcomes of these studies. To obtain meaningful and reproducible results, it is imperative that mechanistic animal model approaches are employed for a more concrete resolution to the involvement of anthranilic acid in depressive disorders.

Contrasting to predicted neurotoxic properties, studies have suggested that anthranilic acid derivatives may have neuroprotective bioactivity [55,185]. Decreased 3-hydroxyanthranilic acid levels in stroke patients, along with the corresponding increase in precursor anthranilic acid, had a negative effect on health outcomes for recovering patients [201]. The inverse, increased 3-hydroxyanthranilic acid levels, is correlated with more signs of recovery in patients that suffered from a stroke [201]. 3-hydroxyanthranilic acid is a redox-active Trp metabolite. The increased levels of this compound in circulation may have a potent anti-inflammatory effect, stemming from both this redox activity and through the redirecting of anthranilic acid towards a more beneficial chemical structure [185,201]. In vitro work with 3-hydroxyanthranilic acid has shown this compound to modulate the pro-inflammatory activities of macrophages, ultimately reducing generalized inflammation [202]. Inflammation plays a known part in many neurological disorders, as highlighted by a recent review [203]. Altogether, this indicates the immunomodulatory activities, and specifically inflammation-regulating activities, of anthranilic acid derivatives which are key parts of this metabolite–host neurological link and a specific disease or etiology.

Research on Trp metabolites and their relation to neurological outcomes is ongoing and has resulted in an abundance of broad descriptive studies for Trp compound relationships in patients with neurological symptoms. Though examples have been identified for strong correlations between anthranilic acid and multiple disease states, not all neurological conditions are showing the same trend. Studies on multiple sclerosis in children and Huntington’s disease showed no or weak correlations with anthranilic acid [204,205]. A recent meta-analysis investigating the presence of Trp-associated compounds across multiple neurological studies showed specific diseases have a weak correlation with anthranilic acid [206]. Results from human–microbiome studies can sometimes be conflicting, underscoring the complexity of host health and metabolic activity research. However, the observation of the Trp and kynurenine metabolic pathway dysregulation as related to host health is well-supported in the literature, but the lack of correlation in cases means this neurological relationship is more complex than a single molecule or single enzymatic reaction. Instead, the relationship for a suite of metabolites to host neurology is likely owed to something more specific regarding the branching and balance (i.e., ‘push/pull’) of these metabolic pathways that are interconnected between the Trp concentration and the co-metabolism of the microbiome and the host.

The key to understanding this apparent paradox between neuroprotective and neurotoxic activity may be in part due to those metabolic pathways which are not linear. These pathways are dynamic networks with many forces directing the movement of metabolism between the arms of complex interconnecting routes and shunts. Anthranilic acid alters the availability of its own primary precursor, kynurenine, and in so doing changes the compound equilibrium. Anthranilic acid is an inhibitor of 3-hydroxyanthranilic acid oxidase (3-HAO) enzymatic activity, the inhibition of which results in decreased production of quinolinic and picolinic acid from 3-hydroxyantranilic acid [55]. Iron is likewise able to inhibit 2-HAO enzymatic activity [50]. Interestingly anthranilic acid can readily chelate this circulating iron, resulting in a complex interaction where iron may indirectly increase anthranilic acid concentrations through 3-HAO inhibition and the increased concentration may then sequester that very same iron [50,55].

In pharmaceutical applications, anthranilic acid and related compounds have exhibited anti-inflammatory properties [158,207]. In a study on rats modeling rheumatoid arthritis, a positive dose-dependent response to synthetic anthranilic acid derivative N-(3′,4′-dimethoxycinnamonyl) anthranilic acid (3,4-DAA) was observed [207]. 3,4-DAA is structurally highly similar with 3-hydroxyanthranilic acid and 3-hydroxykynurenic acid. Arthritic rats received a daily dose of 200, 300, or 400 mg/kg, and all dose groups displayed reduced inflammation via immunomodulatory activity [207]. Endogenous anthranilic acid is also suggested to have these same anti-inflammatory properties [55,208]. 3-hydroxyanthranilic acid also has illustrated anti-inflammatory and antioxidant properties. Such properties are partially derived from the metabolite’s induction of hemooxygenase-1, which is involved in the control of inflammation [185]. As metabolomics work is never black and white, focusing on a single metabolite alone does not provide a clear picture of potential bioactivity. Instead, it is necessary to examine the interplay of network flux, protein regulation, and gene expression, all in combination with an understanding of the host–microbe interplay.

Anthranilic acid, along with its related compounds, cannot be considered singularly neuroprotective or neurotoxic. Perhaps these Trp-related metabolites should be considered more broadly as neuroactive, leaving space for the host context as a driver of ultimate effect relative to the local niche conditions. In the context of human studies, the directionality of metabolic levels cannot be fully assessed. One study reported that increased 3-hydroxyanthranilic acid levels, along with decreased anthranilic acid, were correlated with more beneficial clinic outcomes in stroke patients [201]. Measured serum concentrations post disease diagnosis allows for the theory that many of these measured compounds are the clean-up or response crew rather than the instigators of disease conditions. There is also the additional component of host genetics and the predisposition to certain disease states to consider. Host genetics have been shown to predispose certain individuals to the onset of disease, as seen in some stroke cases [209] and with depressive symptoms [210]. Previous studies have also established that genetic variation across hosts is at least partially responsible, in conjunction with host diet and environment, for driving the composition of the gut microbiota [211,212]. As host genetics play a role in both disease state and microbial composition, it is important to also consider the host genotype in the study of metabolic drivers of disease. Some studies with conflicting metabolic findings may look to the host genotype as one possible explanation for subject-specific differences. To tease apart correlation and causality, along with host-specific findings, future research necessitates the use of tightly controlled studies and the additional use of in vitro models that utilize a time series design to understand the flux and interconversion between the Trp neuroactive catabolites.

## 5. Balancing the Microbes and Metabolites

Metabolic activity is a balancing act between different metabolic routes rather than a singular linear continuation to a compound in a static situation. Much of the current literature discussing bioactive compounds within the same metabolic network nods to this complexity by using ratios rather than absolute concentrations. The concentration of a single compound within a complex metabolic cascade does not often provide the necessary holistic perspective to determine potential health outcomes. Likewise, a singular focus on microorganisms via taxonomic analysis in this metabolic context is likely inadequate to describe a complete picture of such complex microbial interactions and relative abundance changes, even within the gut. Utilizing metabolic ratios, which highlight the push and pull nature of metabolism, in place of static concentrations is one key component for an improved study design. This change enables many complexities to be captured in a dynamic niche that leads considering functional rather than taxonomic location to analyzing microbial consortia. This approach could also be very helpful when considering the microbiome changes to understand what microbes, but also what pathways, are enriching or depleting within the community of metabolism. These combined approaches also enable a quantitative assessment to determine the ranked importance of metabolic routes as well as taxonomic groups as they change in the population relative abundance.

One part of a functional approach to host–microbe–metabolite interactions is looking at the population distribution of Trp-related metabolic genes in bacteria. Considering the push–pull method to regulate this pathway, it is very likely that specific steps in the pathway will be increased or decreased to change the intermediate concentrations as a flux balance. For example, when shunted towards the kynurenine pathway, Trp is first converted to kynurenine via tryptophan indoleamine 2,3-dioxygenase 1 or 2 (IDO1, IDO2) or via the liver enzyme tryptophan 2,3-dioxygenase (TDO) [177]. Kynurenine can then become kynurenic acid, 3-hydroxykynurenine, or anthranilic acid, which pulls the pathway towards other downstream activities [177]. 3-hydroxykynurenine and anthranilic acid both then feed into the production of 3-hydroxyanthranilic acid, the precursor for picolinic acid, quinolinic acid, and NAD+ [177]. A functional protein association network of human IDO1 (Figure 4) [213] confirms the interconnectedness and push–pull nature of the three Trp pathways. A central node of IDO1 shows kynureninase (KYNU), tryptophan 5-monoxygenase (TPH1), and kynurenine formamidase (AFMID) are all in a shared genetic neighborhood (Figure 4A). Gene co-occurrence, as determined using the STRING database [213], reveals a broad distribution of these kynurenine-related enzyme encoding genes across the bacterial kingdom, making the concept of a taxon ratio useful (Figure 4B). Higher resolution down to specific strains will be required to obtain more meaningful insights. Two dominant gut phyla, *Firmicutes* and *Proteobacteria*, both contain species with genes encoding *KYNU*, *TPH1*, *AFMID*. Some *Proteobacteria* also encode *IDO1* but no *Firmicutes* exhibit this ability. A deeper look at *Firmicutes* reveals the distribution of these genes is skewed towards the *Psuedomonadales* order and, interestingly, *Escherichia* lack these four genes. The five *Proteobacteria* orders display a more even distribution of kynurenine enzyme-encoding genes, with *Betaproteobacteria* the only order lacking species encoding *IDO1*. Though the analysis here covers only a small piece of the larger Trp metabolic network, the broad distribution shown here of kynurenine metabolism genes amongst common gut phyla both highlights the complexity of metabolic-related research but also the necessity of a functional approach to such work.

## 6. Conclusions

Host neurology and microbial activity can work in union or conflict with each other, and thus the tight regulation of both is essential for optimal host health. A symbiotic partnership between the gut microbiota and host is a beneficial exchange of substrates that results in increased access to nutrients for the host and readily accessible dietary precursors for the microbes. When the balance shifts, however, the host can experience detrimental neurological effects that are either a result of or exacerbated by microbially produced metabolic products.

Trp is one metabolite whose balance must be carefully regulated. Trp is funneled into the indole, kynurenine, or serotonin pathway, all of which contain other bioactive molecules in their ranks. Some Trp-derived compounds can have positive effects on systemic host functions, like serotonin and the immune and circulatory systems. Other Trp-derived compounds, like quinolinic acid, have detrimental effects on host function and have connections to neurological disorders such as dementia. Another kynurenine-derived bioactive compound produced by both the host and microbe is anthranilic acid. As detailed above, anthranilic acid can be useful as a biomarker for disorders like Alzheimer’s but anthranilic acid may also have a promising future use as a potential therapeutic target for the treatment of devastating and currently incurable neurological diseases. The gut microbiome has the functional capacity to shift the balance of anthranilic acid and subsequent metabolites to the point of disease exacerbation or onset. Controlling this finely tuned metabolic network, especially in the context of the human microbiome, is a challenge that needs additional investigation before being used more practically as biomarkers or therapeutic targets.

## Figures and Tables

**Figure 1 microorganisms-11-01825-f001:**
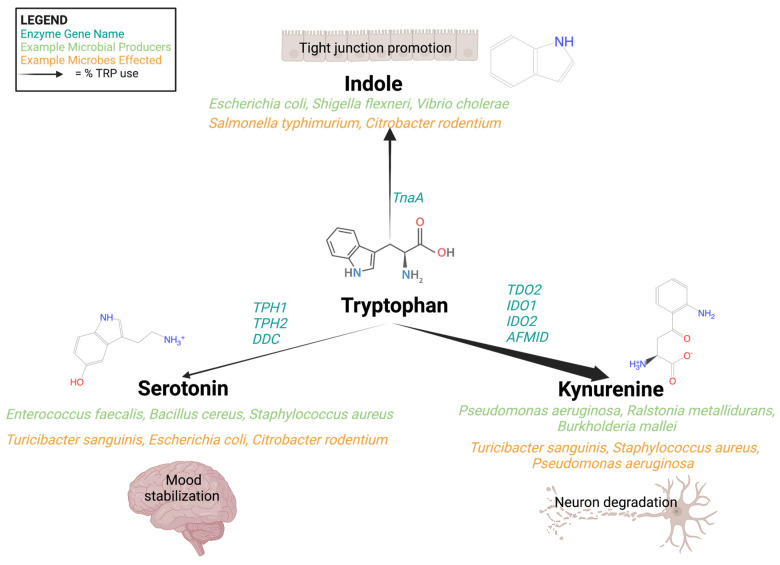
Tryptophan is a central amino acid and is channeled into three pathways: indole, kynurenine, and serotonin. Serotonin and kynurenine are produced by both microbes and mammals via multiple enzymes, listed in teal above. Indole is a tryptophan derivative produced solely by microbes. The majority of tryptophan is moved into the production of kynurenine and related metabolites, with the remaining portion split between indoles and serotonin. Examples of microbial producers of these small molecules are in green while examples of microbes affected by these metabolites are in orange. The broader host effects are graphically represented above.

**Figure 2 microorganisms-11-01825-f002:**
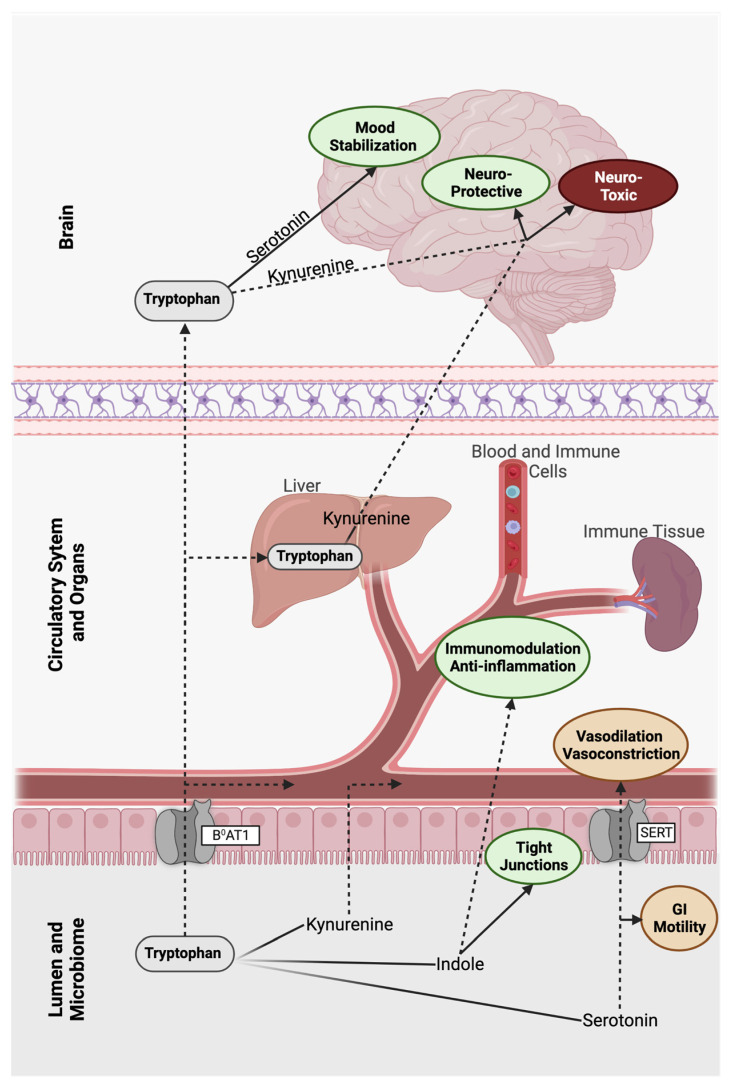
The three major routes of tryptophan after ingestion, metabolism, and absorption in the human gut. Tryptophan can be absorbed into the blood stream and metabolized via the kynurenine pathway in the liver or via the serotonin or kynurenine pathway in the brain. Within the gut, tryptophan can also be funneled towards the kynurenine, indole, or serotonin pathway. These three metabolic pathways contain neuroactive metabolites that affect host physiology and neurology. Fading lines represent metabolism to a primary compound, dotted lines represent metabolism to secondary or downstream metabolites, green text ovals represent positive effects, red ovals represent negative effects, and orange ovals represent neutral outcomes.

**Figure 3 microorganisms-11-01825-f003:**
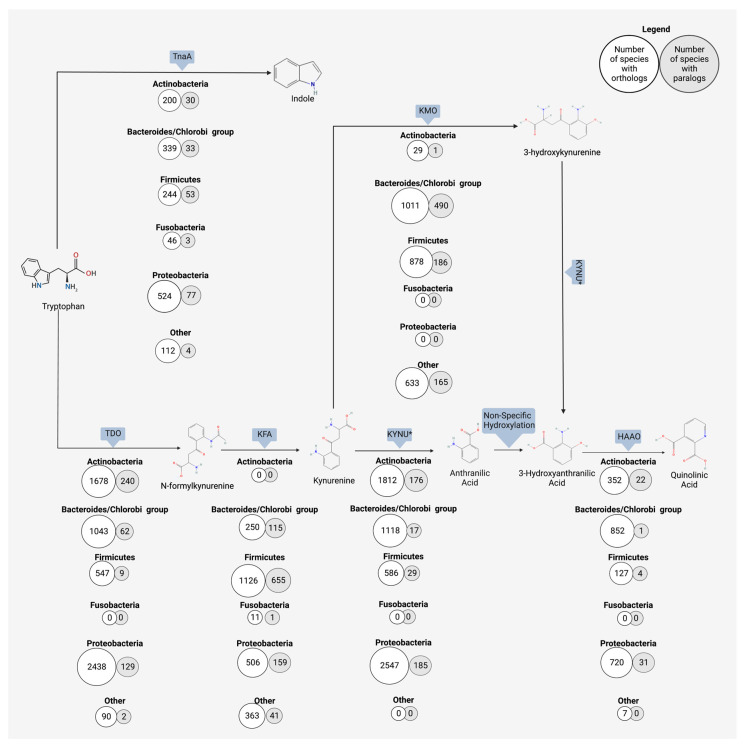
Orthologous and paralogous genes encoding enzymes for the microbial metabolism of tryptophan towards the kynurenine and indole pathways. The number of phyla containing orthologs of the microbial enzyme responsible for each reaction was determined through a search using the OrthoDB database. Enzymes or non-specific reaction types are indicated by rectangular shapes above each phyla list. White circles indicate the number of species in the taxa containing orthologs and gray circles represent the number of species containing paralogs in that phyla. Six chosen phyla are shown, and all other phyla with hits are grouped into “other”. Indole is derived from tryptophan digestion by tryptophanase (TnaA). Down an alternate route from indole, kynurenine can be produced from tryptophan with a two-step process involving tyrptophan 2,3-dioxygenase (TDO) and kynurenine formamidase (KFA). The breakdown of the resulting kynurenine to anthranilic acid by microbes is catalyzed by kynureninase (KNYU). KYNU also catalyzes the reaction of 3-hydroxykynurenine to 3-hydroxyanthranilic acid; this overlap in activity is denoted by an asterisk (*). Anthranilic acid is converted to 3-hydroxyanthranilic acid via non-specific hydroxylation. 3-hydroxyanthranilic acid can then be converted to multiple downstream metabolites, such as picolinic acid (not shown) or quinolinic acid (shown here). The production of quinolinic acid from 3-hydroxyanthranilic acid is catalyzed by 3-hydroxyanthranilate 3,4-dioxygenase (HAAO).

**Figure 4 microorganisms-11-01825-f004:**
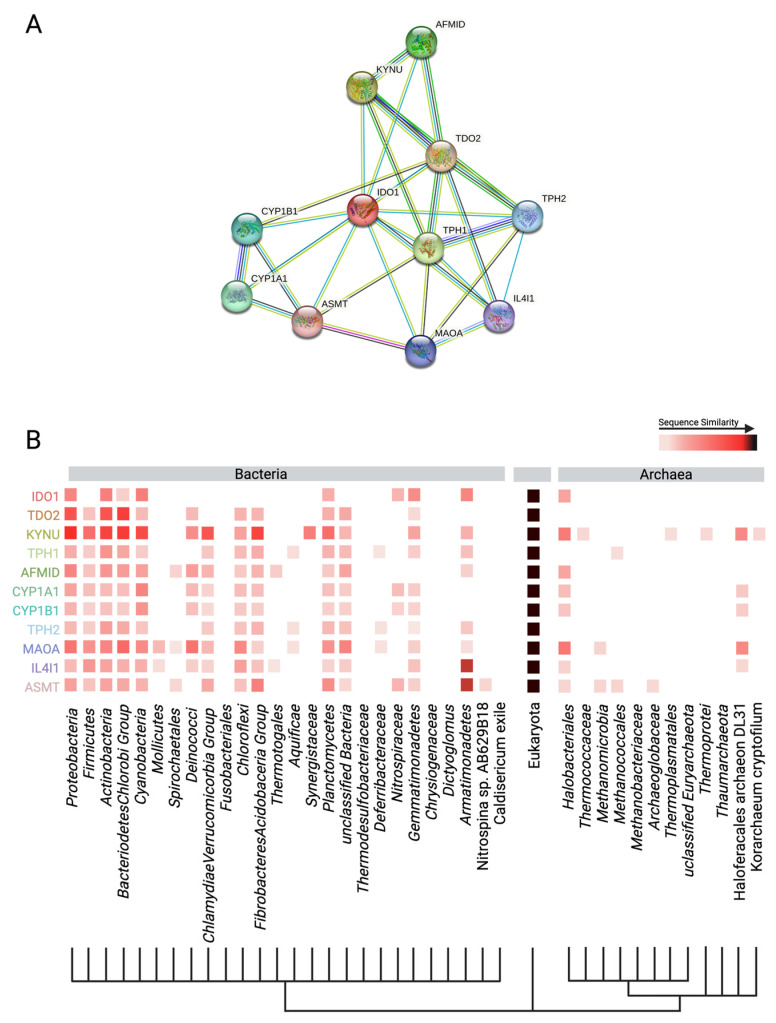
STRING database output from the search for *Escherichia coli* K12 MG1655 indole amine 2,3-dioxygenase 1 (IDO1), one enzyme responsible for the gut conversion of tryptophan to kynurenine. Full protein names are tryptophan 2,3-dioxygenase (TDO2), kynureninase (KYNU), tryptophan hydroxylase (TPH1), kynurenine formamidase (AFMID), human cytochromes P450 1A1 (CYP1A1) P450 1B1 (CYP1B1), tryptophan hydroxylase 2 (TPH2), monoamine oxidase A (MAOA), IL4-induced gene 1 (IL4I1), and acetylserotonin O-methyltransferase (ASMT). (**A**) Protein interaction network with IDO1 as the central node. Light blue connections indicate known interaction from curated databases; pink lines indicate the interaction has been experimentally validated. Light green connections indicate the encoding genes are in the same gene neighborhood; red-orange lines indicate gene fusions, and dark blue indicate gene co-occurrence. Black lines indicate co-expression and light blue indicate protein homology. (**B**) Gene co-occurrence across Bacteria, Eukarya, and Archaea of IDO1- encoding genes and related protein-encoding genes. Color indicates highest sequence similarity within the taxa, with darker reds and black indicating higher similarity.

**Table 1 microorganisms-11-01825-t001:** Tryptophan-derived bioactive metabolites and their functions.

Substrate	Metabolite	Known Microbial Producer (s)	Bioactive Function	Cofactors
Tryptophan	Kynurenine	*Pseudomonas aeruginosa* [42], *Ralstonia metallidurans* [43]	Neuronal damage mediator	INF-gamma [44]
	Indole	*Escherichia coli*, *Proteus vulgaris*, *Clostridium* sp., *Bacteroides* sp. [45]	Anti-inflammatory, epithelial tight-junction regulation	Iron [46]
	Serotonin	*Pseudomonas putida* KT2440 [47]	Mood regulation	Vitamin B2 [48]
Kynurenine	Anthranilic acid	*Pseudomonas aeruginosa* [49], *Burkholderia fungorum*, *Bacillus cereus*, *Ralstonia solanacearum*, and *Ralstonia metallidurans* [43]	Inflammation activation	Iron [50]
	3-Hydroxyanthranilic acid	*Proteobacteria* sp., *Actinobacteria* sp., *Firmicutes* sp. *Bacteroidetes* sp., *Chloroflexi* sp., *Cyanobacteria* sp., *Euryarchaeota* sp. [51]	Free radical generation, apoptotic regulation	Pyridoxal-5′-phosphate [52]
	Kynurenic acid	*Escherichia coli* [53]	NMDA agonist, anticonvulsant	
	3-hydroxykynurenine	*Proteobacteria* sp., *Actinobacteria* sp., *Firmicutes* sp. *Bacteroidetes* sp., *Chloroflexi* sp., *Cyanobacteria* sp., *Euryarchaeota* sp. [51]	Oxidative damage, apoptotic regulation	
	Quinolinic acid	*Streptomyces antibioticus*, *Cyanidium caldarium*, *Karlingia rosea* [54]	NMDA activator	Anthranilic Acid [55]
	Picolinic acid	*Burkholderia xenovorans*, [56]	Iron and zinc chelator, antiviral, antifungal	Anthranilic Acid [55]
Indole	Indole-3-acetamide	Acidobacteria sp., *Bacteroidetes* sp., Firmicutes sp. [57]	Antioxidant, anti-hyperglycemic	
	Tryptamine	*Ruminococcus gnavus* and *Clostridium sporogenes* [58]	Neuroinflammatory modulation	
	Indole-3-acetic acid	*Clostridium* sp., *Bacteroides* sp., *Bifidobacterium* sp. [59]	Anti-angiogenic	
	Indole-3-pyruvic acid	*Pseudomonas fluorescens* and *Candida tropicalis* [60]	Anti-inflammatory, anti-angiogenic	
	Indole-3-acetate	*Clostridium sporogenes* [61]	Anti-inflammatory	
	Indole-3-aldehyde	*Lactobacillus* sp. [62]	Epithelial cell cycle regulation, goblet cell differentiation regulation	
	Indole-3-propionic acid	*Lactobacillus reuteri*, *Akkermansia* sp., *Clostridium* sp., *Peptostreptococi* sp. [63]	Anti-inflammatory, neuronal apoptosis regulation	
	Skatole	*Clostridium* sp. and *Bacteroides* sp. [64]	Induces colonocyte apoptosis	
Serotonin	Melatonin	*Roseburia hominis* [65]	Circadian rhythm regulation, antioxidant	Tetrahydrobiop-terin and O_2_ [66]

## Data Availability

The STRING database can be accessed at https://string-db.org and Ortho DB can be accessed at https://orthodb.org.

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
