# Peer review of "Microbial-Derived Tryptophan Metabolites and Their Role in Neurological Disease: Anthranilic Acid and Anthranilic Acid Derivatives"

_microorganisms, 2023, doi:10.3390/microorganisms11071825_

Round 1
Reviewer 1 Report
Dear Authors,
Please find the following corrections:
- Please correct the word “derivates” with “derivatives” in the title (line 4) and throughout the whole text (lines 167, 285, 341, and 557).
- In the introduction, please authenticate the first paragraph with reference. It is also possible to discuss the scientific view (lines 134-158) considering related previous work.
- The data represented in Table 1, Figure 1, and Figure 3 are not readable, please clarify the font.
Author Response
Response to Reviewers
Dear Editor and Reviewers,
Thank you for the time you took to read our paper and likewise for your thoughtful responses. We have made the requested edits, as seen in the main text and below.
Reviewer 1
Comment 1: Please correct the word “derivates” with “derivatives” in the title (line 4) and throughout the whole text (lines 167, 285, 341, and 557).
Thank you for catching these misspellings. We have corrected them throughout the document.
Line 166: “The objective of this narrative review is to highlight the importance of anthranilic acid and its derivatives by outlining the current understanding of anthranilic acid’s role in health and disease, followed by a discussion of the microbes that may affect the production and circulation of anthranilic acid.”
Line 285: “Trp, regardless of the source, is central to many cellular functions and the general path from Trp to bioactive derivatives is important study for host health and disease impacts.”
Line 337: “Not all microbes can produce indole. Fungi and bacteria that are unable to synthesis indole or its derivatives still have mechanisms for sensing and utilizing indoles as metabolic precursors or ligands for signaling cascades [103, 104].”
Line 563: “A 2021 review of anthranilic acid medicinal research highlights the bioactivity of anthranilic acid derivatives [158].”
Comment 2: In the introduction, please authenticate the first paragraph with reference. It is also possible to discuss the scientific view (lines 134-158) considering related previous work.
Thank you for noting that our introduction could use more references. We have now added references into the first and last paragraphs of the introduction. Please see the new citations below:
Line 37-38: “This difficulty is especially pronounced in the human gut, where extensive overlap between small molecule metabolism of the host and microbiome exists [1].”
Line 42-44: “In the broader context of host health, the importance of the host GT activity and resident microbiome contributions cannot be overstated, specifically with the extensive reach of small molecules to distant tissues in the body [2].”
Line 138-139: “Additionally, host diet, the source of substrates for the microbiome, may also be in flux [29].”
Line 139-141: “Many studies examining the gut microbiome focus primarily on taxonomic diversity, and do not make the link to metabolism and metabolite changes that are inevitably from diet and host metabolism [30, 31].”
Line 141-143: “While a necessary part of the puzzle, bacterial identity alone does not indicate microbiome functionality, which changes the collective metabolism and metabolome [32-35].”
Comment 3: The data represented in Table 1, Figure 1, and Figure 3 are not readable, please clarify the font
Table 1, Figure 1, and Figure 3 all now have larger and easier to read font. The updated table and figures can be found in the revised main text.

Reviewer 2 Report
This a thorough narrative review on a hot topic in research, that is, tryptophan catabolism. The Authors have synthesized loads of evidence microbial-derived tryptophan metabolites and their role in neurological diseases, highlighting the most relevant findings in this field in a pleasant-to-read way.
I have no critical concerns regarding this submission, yet I have some suggestions for the Authors:
1. The fact that this is a narrative, non-systematic review should be clearly stated in title, abstract, and text body (lines 160-169).
2. When dealing with this topic, some fundamental papers absolutely need to be cited: e.g., Schwarcz et al., 2012, Nat Rev Neurosci. https://doi.org/10.1038/nrn3257.
3. Amongst psychiatric disorders in which tryptophan catabolism is altered, besides schizophrenia and major depressive disorder, it is very important that the Authors also mention bipolar disorder and the role of kynurenine pathway alterations in this [https://doi.org/10.1038/s41380-020-00913-1; https://doi.org/10.1192/j.eurpsy.2022.2340].
Author Response
Response to Reviewers
Dear Editor and Reviewers,
Thank you for the time you took to read our paper and likewise for your thoughtful responses. We have made the requested edits, as seen in the main text and below.
Reviewer 2
This a thorough narrative review on a hot topic in research, that is, tryptophan catabolism. The Authors have synthesized loads of evidence microbial-derived tryptophan metabolites and their role in neurological diseases, highlighting the most relevant findings in this field in a pleasant-to-read way.
I have no critical concerns regarding this submission, yet I have some suggestions for the Authors:
Comment 1: The fact that this is a narrative, non-systematic review should be clearly stated in title, abstract, and text body (lines 160-169).
We have made it clearer that this is a narrative review by specifying that fact in line 26 and line 166. As the title does not explicitly say “review” we would like to leave the title unchanged.
Line 26: “This narrative review provides an overview of the current understanding of anthranilic acid’s role in neurological health and disease, with particular focus on the contribution of the gut microbiome, the gut-brain axis, and the involvement of the three major tryptophan pathways.”
Line 166: “The objective of this narrative review is to highlight the importance of anthranilic acid and its derivatives by outlining the current understanding of anthranilic acid’s role in health and disease, followed by a discussion of the microbes that may affect the production and circulation of anthranilic acid.”
Comment 2: When dealing with this topic, some fundamental papers absolutely need to be cited: e.g., Schwarcz et al., 2012, Nat Rev Neurosci. https://doi.org/10.1038/nrn3257.
Thank you for highlighting some key literature we could cite that improves the overall paper. The recommended Schwarcz et al. paper [28] has now been included as one of the primary cited pieces in the introduction section of the review.
Line123-126: “The kynurenine pathway, one of three main Trp metabolic pathways, has garnered increasing interest for its connection to neurological diseases through the production of the bioactive compounds quinolinic, picolinic, kynurenic and anthranilic acid in this pathway [21-28].”
Comment 3: Amongst psychiatric disorders in which tryptophan catabolism is altered, besides schizophrenia and major depressive disorder, it is very important that the Authors also mention bipolar disorder and the role of kynurenine pathway alterations in this [https://doi.org/10.1038/s41380-020-00913-1; https://doi.org/10.1192/j.eurpsy.2022.2340].
Like the above comment, we appreciate the suggestion for further literature that rounds out the review. The two recommended Bartoli et al papers have been included in the review and the excerpts can be found below:
Line 532-529: “Kynurenine metabolites have also been linked to bipolar disorder, which is a complex neurological disorder that affects approximately 2% of the population [150, 151]. Notably, recent meta-analyses have shown there is a connection between metabolites of the kynurenine pathway and bipolar disorder, but the mechanistic and possibly causative relationship remains less well understood [150, 151]. Though the causative relationship may not yet be fully elucidated, it is nevertheless clear that metabolites stemming from the kynurenine pathway play key roles as markers of disease or perhaps even causative agents.”
